# Structural basis for plant plasma membrane protein dynamics and organization into functional nanodomains

Julien Gronnier[1], Jean-Marc Crowet[2†], Birgit Habenstein[3†], Mehmet Nail Nasir[2†], Vincent Bayle[4†], Eric Hosy[5], Matthieu Pierre Platre[4], Paul Gouguet[1], Sylvain Raffaele[6], Denis Martinez[3], Axelle Grelard[3], Antoine Loquet[3], Françoise Simon-Plas[7], Patricia Gerbeau-Pissot[7], Christophe Der[7], Emmanuelle M Bayer[1], Yvon Jaillais[4], Magali Deleu[2], Véronique Germain[1], Laurence Lins[2*], Sébastien Mongrand[1*]

[1]Laboratoire de Biogenèse Membranaire (LBM), Unité Mixte de Recherche UMR 5200, CNRS, Université de Bordeaux, Bordeaux, France; [2]Laboratoire de Biophysique Moléculaire aux Interfaces, GX ABT, Université de Liège, Gembloux, Belgium; [3]Institute of Chemistry and Biology of Membranes and Nanoobjects (UMR5248 CBMN), CNRS, Université de Bordeaux, Institut Polytechnique Bordeaux, Pessac, France; [4]Laboratoire Reproduction et Développement des Plantes, Université de Lyon, ENS de Lyon, Université Claude Bernard Lyon 1, Lyon, France; [5]Interdisciplinary Institute for Neuroscience, CNRS, University of Bordeaux, Bordeaux, France; [6]LIPM, Université de Toulouse, INRA, CNRS, Castanet-Tolosan, France; [7]Agroécologie, AgroSup Dijon, INRA, Université Bourgogne Franche-Comté, F-21000 Dijon, ERL 6003 CNRS, Dijon, France

*For correspondence:
l.lins@ulg.ac.be (LL);
sebastien.mongrand@u-bordeaux.
fr (SM)

†These authors contributed
equally to this work

Competing interests: The
authors declare that no
competing interests exist.

Reviewing editor: Gary Stacey,
University of Missouri, United
States

**Abstract** Plasma Membrane is the primary structure for adjusting to ever changing conditions. PM sub-compartmentalization in domains is thought to orchestrate signaling. Yet, mechanisms governing membrane organization are mostly uncharacterized. The plant-specific REMORINs are proteins regulating hormonal crosstalk and host invasion. REMs are the best-characterized nanodomain markers via an uncharacterized moiety called REMORIN C-terminal Anchor. By coupling biophysical methods, super-resolution microscopy and physiology, we decipher an original mechanism regulating the dynamic and organization of nanodomains. We showed that targeting of REMORINis independent of the COP-II-dependent secretory pathway and mediated by PI4P and sterol. REM-CA is an unconventional lipid-binding motif that confers nanodomain organization. Analyzes of REM-CA mutants by single particle tracking demonstrate that mobility and supramolecular organization are critical for immunity. This study provides a unique mechanistic insight into how the tight control of spatial segregation is critical in the definition of PM domain necessary to support biological function.

## Introduction

Membrane proteins and lipids are dynamically organized in domains or compartments. Emerging evidences suggest that membrane compartmentalization is critical for cell signaling and therefore for development and survival of organisms ( *Grecco et al., 2011* ). The understanding of molecular mechanisms governing protein sub-compartmentalization in living cells is one of the most critical issues regarding the understanding of how membranes function.

In this paper, we exploit the protein family REMORIN ( REMs ), the best-characterized PM-domain markers in plants ( *Raffaele et al., 2009* ; *Jarsch et al., 2014* ; *Jarsch and Ott, 2011* ). REMs belong to a multigenic family of six groups encoding plant-specific membrane-bound proteins involved in responses to biotic and abiotic stimuli ( *Raffaele et al., 2009* ; *Jarsch et al., 2014* ; *Jarsch and Ott, 2011* ; *Raffaele et al., 2007* ; *Gui et al., 2014* ; *Jamann et al., 2016* ).The physiological functions of REMs have been poorly characterized. To date, their involvement has been clearly reported in plant - microbe interactions and hormonal crosstalk: in *Solanaceae* , group 1 REMs limit the spreading of Potato Virus X ( PVX ), without affecting viral replication ( *Raffaele et al., 2009* ), and promote susceptibility to *Phytophthora infestans* ( *Bozkurt et al., 2014* ). A group 2 REM was described as essential during nodulation process in *Medicago truncatula* ( *Lefebvre et al., 2010* ; *Tóth et al., 2012* ).In rice, a group 4 REM is upregulated by abscissic acid and negatively regulates brassinosteroid signaling output ( *Gui et al., 2016* ). Arabidopsis group 4 REMs also play a role as positive regulators of geminiviral infection ( *Son et al., 2014* ). Finally, group 1 and group 6 REMs regulate the PM-lined cytoplasmic channels called plasmodesmata ( PD ), specialized nanochannels allowing intercellular communication in plants. Remarkably, these latter isoforms of REMs were found to be able to modify PD aperture leading to a modification of viral movement ( *Raffaele et al., 2009* ; *Perraki et al., 2014*) and an impact on the grain setting in rice, respectively ( *Gui et al., 2014* ). To fulfill these functions, REMs need to localize to the PM ( *Perraki et al., 2012* ; *Bozkurt et al., 2014* ). Nevertheless, the functional relevance of REM PM-nanodomain organization and the molecular mechanisms underlying PM-nanodomain organization of REM remain to be elucidated.

Our groups defined a short peptide located at the C-terminal domain of REM, called REM-CA (REM C-terminal Anchor) as a novel membrane-binding domain shaped by convergent evolution among unrelated putative PM-binding domains in bacterial, viral and animal proteins. ( *Perraki et al., 2012* ; *Raffaele et al., 2013* ; *Konrad et al., 2014* ). REM-CA is necessary for PM localization of REMs and sufficient to target a given soluble protein ( eg GFP ) to the PM, highlighting that PM-domain localization is conferred by the intrinsic properties of REM-CA. REM-CA binds in vitro to poly phosphoinositidesbut the association of REM-CA with the PM is not limited to electrostatic interactions and the final interaction with PM display anchor properties similar to intrinsic protein ( *Perraki et al., 2012* ). REM-CA can be S-acylated ( *Hemsley et al., 2013* ), although this modification is not the determinant for their localization to membrane domains, since deacylated versions of some REMs remain able to localize to membrane domains ( *Jarsch and Ott, 2011* ; *Hemsley et al., 2013* ). The specificity of targeting, the anchoring and nanoclustering mechanisms mediated by REM-CA to the PM inner-leaflet nanodomains remain therefore elusive.

In this paper, using various modeling, biophysical, high-resolution microscopy and biological approaches, we deciphered an original and unconventional molecular mechanism of REM anchoring to PM: the target from the cytosol to PM by a specific PI4P-protein interaction, a subsequent folding of the REM-CA in the lipid bilayer, and its stabilization inside the inner-leaflet of the PM leading to an anchor indistinguishable from an intrinsic membrane protein. By constructing mutants, we were able to alter REM PM-nanodomain organization. Unexpectedly, single molecule studies of REM mutants reveal that single-molecule mobility behavior is not coupled to supramolecular organization. These mutants were unable to play their role in the regulation of cell-to-cell communication and plant immune defense against viral propagation,

## Results and discussion

To specifically analyze the implication of REM-CA-lipid interactions in membrane targeting, we used the naturally non-S-acylated REM variant: *Solanum tuberosum* REMORIN group 1 isoform three called *St* REM1.3 ( *Figure 1—figure supplement 1*). *St* REM1.3 is the best-studied isoform of REMs: *St* REM1.3 is a trimeric protein ( *Perraki et al., 2012* ) that strictly localizes to the PM and segregates in a sterol-dependent manner into ca. 100 nm nanodomains ( *Raffaele et al., 2009* ; *Demir et al., 2013* ). In this study, we use *Nicotinana benthamiana*leaf epidermal cells as a model tissue. In this context, we showed that *St* REM1.3 is a functional homolog of the PM-localized *Nicotinana benthamiana* endogenous group 1b REMs toward the restriction of Potato Virus X ( PVX ) spreading. Consistently, *Nb* REM1.2 and *Nb* REM1.3 isoforms are highly expressed in leaf epidermis (*Figure 1—figure supplement 2*). Importantly, PVX is a mechanically transmitted virus for which the infection initiates in the epidermis and spreads from cell-to-cell via different tissues to reach the phloem

vasculature and infect the whole plant ( *Cruz et al., 1998*). In this context, leaf epidermis is the appropriate tissue to study the role of PM-nanodomains in the cell's responses to viruses.

## *St* REM1.3 is targeted to PM domains by a mechanism likely independent of the secretory pathway

Trafficking studies of REMs in plant cells showed that their PM localization (observed as secant or tangential views of epidermal cells, *Figure 1A* ) seem not rely on vesicular trafficking ( *Raffaele et al., 2013* ; *Gui et al., 2015* ; *Konrad et al., 2014* ). Consistent to what was shown for the rice group 4 REM ( *Gui et al., 2015* ), YFP- *St* REM1.3 was normally targeted to the PM upon inhibition of COP-II-mediated ER-to-Golgi trafficking by overexpression of a dominant-negative SAR1 ( *de Marcos Lousa et al., 2016* ) or by treatment with Brefeldin A ( BFA), a pharmacological inhibitor of ADP-ribosylation factor 1-GTPase and its effectors, ARF-guanine-exchange factors, of the COP-I-mediated secretory pathway ( *Peyroche et al., 1999* ) ( *Figure 1B* and *Figure 1—figure supplement 3* ). Moreover, organization of YFP- *St* REM1.3 in the plane of the PM was not affected in the presence of dominant-negative SAR1 ( *Figure 1B* ) as quantified by the Spatial Clustering Index ( SCI ) calculated as the max-to-min ratio of fluorescence intensity in the PM ( *Figure 1—Figure supplement 4* ).

*St*REM1.3 being a hydrosoluble protein intrinsically attached to the PM with no transit peptide, no transmembrane domain and no membrane anchor signatures (*Raffaele et al., 2007*), altogether our data suggest that *St*REM1.3 (likely under the form of a trimer [*Perraki et al., 2012*]) is targeted to the PM from the cytosol by a mechanism independent of the classical COP-II-mediated secretory pathway and that the formation of *St*REM1.3 PM-nanodomains likely does not rely on secretory trafficking.

## REMORIN localization into highly-ordered PM-nanodomains is mediated by sterol and phosphatidylinositol 4-phosphate

Group 1 REMs co-purify in the detergent-resistant membrane biochemical fraction with sterols and phosphoinositides (PIPs) (*Raffaele et al., 2009*; *Demir et al., 2013*; *Mongrand et al., 2004*; *Furt et al., 2010*). REM-CA also binds to PIPs in vitro (*Perraki et al., 2012*). We therefore tested the involvement of both sterols and PIPs in the PM-nanodomain localization of *St*REM1.3 in vivo by modifying the PM lipid content. First, to alter membrane sterol composition we chose fenpropimorph (fen). Fen alters the PM sterol-composition but not the total amount of sterols (*Grison et al., 2015*; *Hartmann et al., 2002*). As expected, after fen treatment, the content of $\Delta 5$-sterols decreased concurrently with an increase in cycloartenol (*Figure 1—figure supplement 5A*). This qualitative modification of sterol composition had no effect on the targeting of YFP-*St*REM1.3 to the PM but abolished its nanodomain organization measured by the SCI (*Figure 1C*). Proton pump PMA fused to GFP was used as control of membrane integrity after fen treatment (*Figure 1—figure supplement 5B–E*). To obtain further evidence of the enrichment of *St*REM1.3 into sterol-enriched nanodomains, expected to display a higher degree of order (*Dufourc, 2008*), we used the environment–sensitive probe di-4-ANEPPDHQ in vivo (*Zhao et al., 2015*). *Figure 1D* shows that nanodomains enriched in YFP-*St*REM1.3 co-localized with highly ordered regions of the PM, in good agreement with the involvement of sterols in *St*REM1.3 localization.

Second, we tested the implication of PIPs in *St* REM1.3 recruitment to the PM. We focused in particular on Phosphatidylinositol 4-phosphate (PI4P). Recent works showed that PI4P is enriched in the inner-leaflet of plant PM, conferring a negatively charged electrostatic field that defines PM identity in regard to other endomembranes ( *Vermeer et al., 2009* ; *Simon et al., 2016* , *Simon et al., 2014* ). To alter the PI4P content, we used the Myristoylated / Palmitoylated-Phosphatidylinositol 4-phosphatase SAC1p enzyme from yeast ( *Hammond et al., 2012* ) fused to mTurquoise2 (MAP-mTU2-SAC1p) which specifically dephosphorylates PI4P at the PM level ( *Hammond et al., 2012Hammond et al., 2012* ) *Hammond et al., 2012* ; *Stefan et al., 2011*; *Mesmin et al., 2013* ; *Moser von Filseck et al., 2015a* , *Moser von Filseck et al., 2015b* ) without impacting PS or PI (4,5) P $_2$ ( *Simon et al., 2016* ) ( *Figure 1—figure supplement 6* ). Compared to the expression of the dead version of MAP-SAC1p, the expression of the active form induced a reduction in PM-associated PI4P concentration leading to a strong decrease of both YFP- *St* REM1.3 signal and lateral

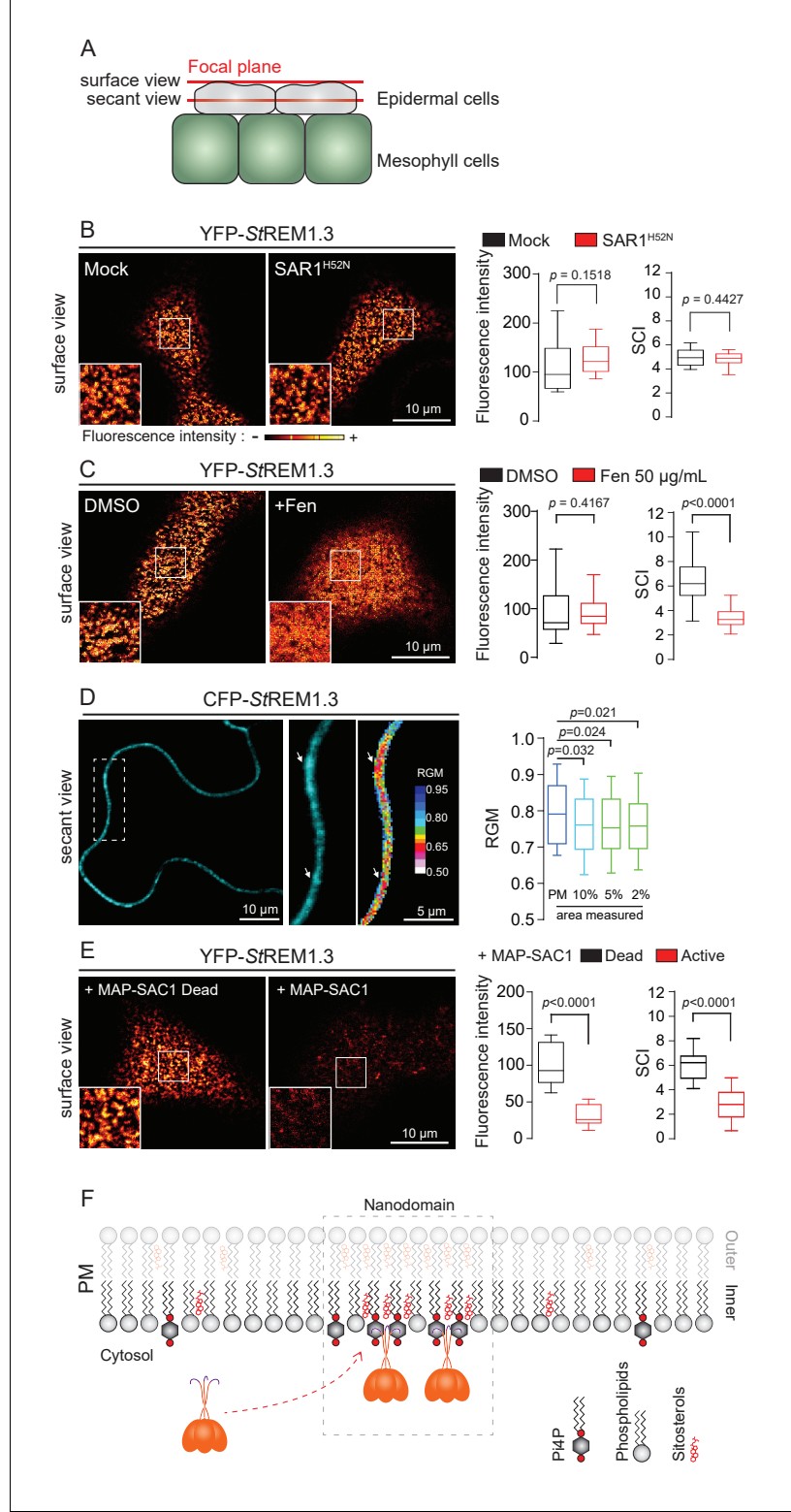

**Figure 1.** REMORIN localization into highly ordered PM nanodomains is mediated by sterols and PI4P. (**A**) Explanatory schematic of the secant or surface views of *N. benthamiana* leaf abaxial epidermal cell plasma membrane (PM) used throughout the article. (**B**) Confocal imaging surface views of *Nicotinana benthamiana* leaf epidermal cells expressing YFP-*St*REM1.3 with or without dominant-negative SAR1[H52N] (PMA4-GFP was used as a potency control, see *Figure 1—figure supplement 2*), 24 hr after agroinfiltration. Tukey boxplots show the
*Figure 1 continued on next page*

*Figure 1 continued*

mean fluorescence intensity and the Spatial Clustering Index , SCI ( n = 3, quantification made on a representative experiment, at least 38 cells per condition). ( C ) Surface view confocal images showing the effect of Fenpropimorph (Fen) on PM patterning of YFP- *St* REM1.3 domains 20 hr after agroinfiltration. Tukey boxplots show the mean fluorescence intensity and the SCI of YFP- *St* REM1.3 in the Mock ( DMSO ) or Fen-treated leaves ( 50 μg / mL ), at least 46 cells from three independent experiments. ( D ) Secant view confocal fluorescence microscopy images displaying the degree of order of CFP- *St*REM1.3 - enriched domains (left panel) by the environment - sensitive probe Di - 4 - ANEPPDHQ (middle panel) 48 hr after agroinfiltration. Di - 4 - ANEPPDHQ red / green ratio ( RGM ) was measured for the global PM, and for the 10, 5, 2 % most intense CFP- *St* REM1.3 signal-associated pixels (right panel). A lower red / green ratio is associated with an increase in the global level of membrane order, at least 70 cells from three independent experiments. ( E ) Surface view confocal images showing the effect of dead or active constructs of MAP-SAC1p (MAP-mTurquoise2-SAC1p from yeast, see *Figure 1—figure supplement 5* ) on PM domain localization of YFP- *St* REM1.3 20 hr after agroinfiltration. Tukey boxplots show the mean fluorescence intensity and the SCI of YFP- *St* REM1.3, at least 52 cells from four independent experiments. ( F ) Model showing the PI4P-driven targeting of the trimer of *St* REM1.3 to the PM and its PI4P- and sterol-dependent nanodomains organization. In all panels, p - values were determined by a two-tailed Mann-Whitney test.

The online version of this article includes the following figure supplement(s) for figure 1:

**Figure supplement 1.** Sequence alignment of 51 group 1 REMORIN C-terminal Anchor sequences.
**Figure supplement 2.** *Nicotiana benthamiana* Group 1b REMORINs are expressed in leaf epidermal cells, encode for PM nanodomain localized proteins and are functional homologs of StREM1.3 toward PVX propagation.
**Figure supplement 3.** YFP- *St* REM1.3 is targeted to the PM-domains by a mechanism independent of the COP-I / COP-II secretory pathway.
**Figure supplement 4.** Spatial clustering index calculated as the max-to-min ratio of fluorescence intensity in the PM.
**Figure supplement 5.** Modification of the sterol pool of *N. benthamiana* leaves by the drug Fenproprimorph (fen).
**Figure supplement 6.** Myristoylation and Palmitoylation ( MAP ) -mTurquoise2-SAC1p localizes at PM of *N. benthamiana* leaf epidermal cells and specifically depletes PM PI4P but not PI (4,5) P $_2$ or PS.

segregation at the PM ( *Figure 1E* ). These data suggest that PI4P is required for the targeting of *St* REM1.3 at the PM and for its sub-compartmentalization within the PM plane.

Altogether, these results suggest that PM inner-leaflet lipids, notably sterols and PI4P are critical for the targeting of the *St* REM1.3 to PM nanodomains by a mechanism independent of the classical secretory pathway ( *Figure 1F* ).

## REM-C-terminal anchor peptide is an unconventional PM-binding domain embedded in the bilayer that folds upon specific lipid interaction

As mentioned before, REM-CA is critical for PM-targeting ( *Perraki et al., 2012* ; *Raffaele et al., 2013* ; *Konrad et al., 2014* ; *Gui et al., 2015* ) ( *Figure 2A* ). To better understand the role of lipids and the function of REM-CA in the assembly of *St* REM1.3 into nanodomains, we used a combination of biophysical, modeling and biological approaches.

First, liquid-state NMR spectra of REM-CA in aqueous environment showed that REM-CA peptide is unstructured ( *Perraki et al., 2012* ) ( *Figure 2—Figure supplement 1A* ). Second, equivalent spectra, acquired in hydrophobic environment showed that REM-CA folds into an alpha helical conformation ( *Figure 2—Figure supplement 1A* ). Third, to gain insights into the embedment of REM-CA in the PM we performed solid-state NMR experiments on liposomes mimicking the PM inner-leaflet composition ie containing phosphatidylcholine ( PC ), PIPs and phosphatidylserine ( PS ) ( *Cacas et al., 2016* ) ( *Figure 2—figure supplement 1B, C* shows the lipid content of the phosphoinositide mix, further called PIPs, used in this study). In these conditions, REM-CA's partial insertion into liposomes increased the degree of order of the first 10 carbon atoms of acyl chains indicated that REM-CA is partially embedded in the lipid phase ( *Figure 2B* ). Importantly, REM-CA insertion does not modify the overall bilayer structure as suggested by a very similar global thermotropism ( *Figure 2—figure supplement 1D, E* ).

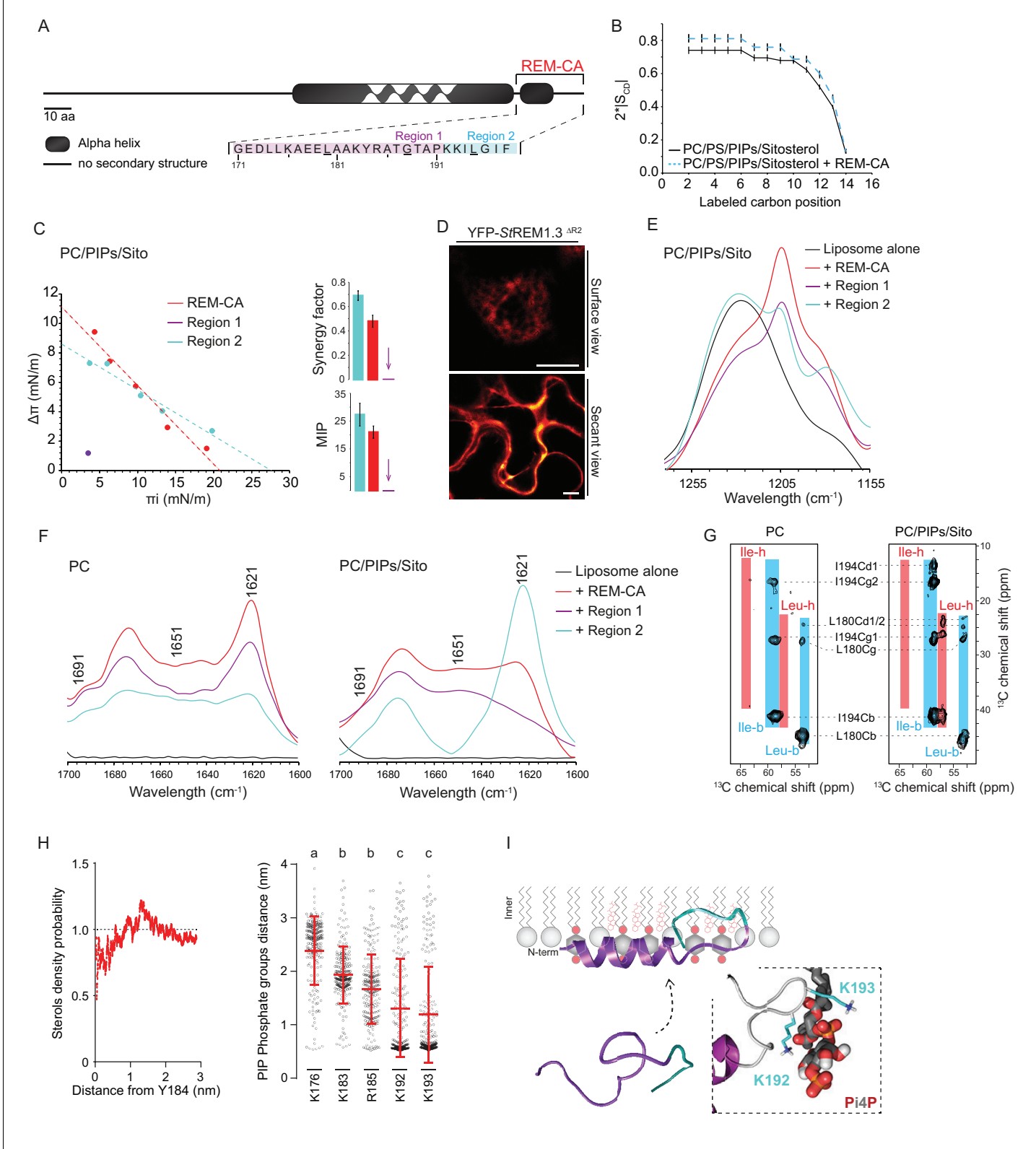

**Figure 2.** | REMORIN C-terminal anchor peptide is an unconventional PM-binding domain embedded in the bilayer that folds upon specific lipid interaction. (A) Primary sequence of *St*REM1.3 showing the two putative regions 1 and 2 (R1 and R2) composing the REM-CA. Hatched domain represents the putative coiled-coil helix. (B) Order parameter of the carbon atoms of the fatty acid moiety of all-deuterated 1,2-dimyristoyl-D54-*sn*-3-phosphocholine (DMPC-d54) in PM inner leaflet-mimicking liposomes revealed by deuterium NMR. (C) Plots of the maximal surface pressure variation

*Figure 2 continued on next page*

*Figure 2 continued*

(ΔΠ) vs. the initial surface pressure (Πi) (left panel) and the corresponding maximal insertion pressure (MIP) and synergy factor (right panel) obtained from the adsorption experiments performed viaa Langmuir trough with a monolayer composed of phosphatidylcholine (PC), phosphoinositides (PIPs) and sitosterol (Sito) (see *Figure 2—figure supplement 3A*). The insignificant ΔΠ obtained for D1 indicates that D1 cannot penetrate into the monolayer. (D) Subcellular localization of YFP-*St*REM1.3 deleted for R2, transiently expressed in *N. benthamiana* leaf epidermal cells. Scale bars, 10 μm. (E) FT-IR spectra measured in the 1155–1255 cm$^{-1}$ absorbance region for the REM-CA, R1 and R2 peptides inserted into MLVs composed of PC:PIPs: Sito (see *Figure 2—figure supplement 3B*). (F) FT-IR spectra in the 1600–1700 cm$^{-1}$ absorbance region for the REM-CA, R1, R2 peptides and liposome alone with MLVs composed of PC alone and PC:PIPs:Sito (see *Figure 2—figure supplement 3B*). (G) Solid-state NMR spectra of REM-CA peptides co-solubilized with DMPC-d54 supplemented with PIPs and Sitosterol (see *Figure 2—figure supplement 3C*). Excerpts on the position of the Cα resonance frequencies of Leucines and Isoleucines on the abscissa are depicted. (H) Radial distribution functions (RDF) of Y184 and sterols, and average distances between the five lysine(K)/arginine(R) residues of REM-CA and the phosphate groups of PI4P during MD simulation, bar indicates mean ± s.d., letters indicate significant differences revealed by Dunn's multiple comparisons test p<0.0001. (I) Model of the insertion of REM-CA in the PM inner-leaflet based on tensiometry, FTIR, MD and NMR studies. Inset displays Molecular Dynamics (MD) model of the two lysines, K192 and K193, likely in interaction with the phosphate groups of PIPs.

The online version of this article includes the following figure supplement(s) for figure 2:

**Figure supplement 1.** Solution NMR and $^{31}$P and $^{2}$H solid-state NMR analysis.

**Figure supplement 2.** In silico analysis of REM-CA from *St* REM1.3 suggests the existence of two distinct structural regions.

**Figure supplement 3.** Biophysical studies evidence of the interaction of REM-CA with lipids.

**Figure supplement 4.** Molecular dynamics ( MD ) simulation reveals interactions between REM-CA residues and lipids in the ternary lipid mixture.

Next, we determined the REM-CA peptide regions that are inserted in the hydrophobic core of the bilayer. In silico analyzes predicted that REM-CA is structurally divided into two regions ( *Figure 2—figure supplement 2* ): a putative helical region (171-190aa, called Region 1, R1) and a more hydrophobic non-helical region (191-198aa called Region 2, R2). We thus tested the ability of REM-CA, R1 or R2 peptides alone to insert into monolayers mimicking the PM inner-leaflet ( *Cacas et al., 2016* ). Adsorption assays showed that the penetration capacity of the peptide REM-CA was higher in the monolayers composed of PC, PIPs and sitosterols, than in monolayers composed of PC alone ( *Figure 2—figure supplement 3A*). Furthermore, peptide R2 but not the R1 was able to insert into monolayers ( *Figure 2C* ). Consistently, deletion of R2 in the REM-CA of YFP- *St* REM1.3 abolished PM association *in planta* ( *Figure 2D* ).

We next performed Fourier transform-infrared spectroscopy (FT-IR) to characterize REM-CA-lipid interactions at atomistic level. FT-IR showed a maximum intensity shift in the absorbance wave-number of lipid phosphate groups in the presence of REM-CA, R1 and R2 peptides, with a stronger effect of R1 as compared to R2 ( *Figure 2—figure supplement 3B* ). This clearly shows that the polar heads of lipids are involved in the REM-CA-liposome interactions. Moreover, a maximum intensity shift in the absorbance wavenumber of carbon-hydrogen bonds was observed in presence of R2, which confirmed that R2 is more embedded than R1 within the lipid phase ( *Figure 2E* ).

To further inquire into the role of lipids in the folding of REM-CA, we performed structural analyses by FT-IR and solid-state NMR of the peptides in liposomes containing either PC alone, or PC with PIPs and sitosterols. FT-IR experiments showed that REM-CA, R1 and R2 peptides are mainly a mix of different structures in PC-containing liposomes (*Figure 2F*). In contrast, R1 was more helical and R2 was more extended when sitosterol and PIPs are present in the bilayer (*Figure 2F*). Importantly, PIPs were

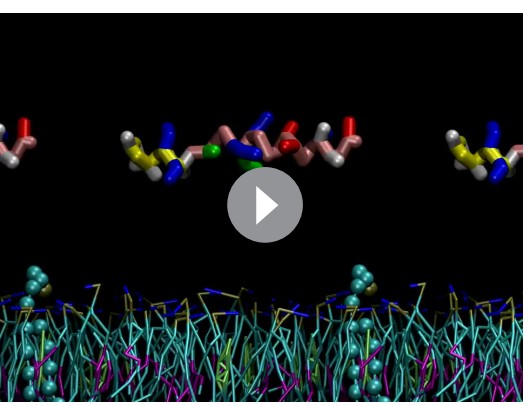

**Video 1.** Molecular dynamics (MD) simulation reveals interactions between REM-CA residues and lipids in the ternary lipid mixture. MD coarse-grained simulations propose a model of the insertion of REM-CA in the lipid bilayer (PLPC/PI4P/sitosterol), where peptide-lipid interaction would be mediated by the interaction of REM-CA with the phosphate groups of PI4P (in gold) and the embedment of the region 2 (in yellow) inside the lipid phase, see online methods.
https://elifesciences.org/articles/26404#video1

sufficient to induce R1, R2 and REM-CA folding (*Figure 2—figure supplement 3B*).

Lipid-mediated folding of REM-CA, embedded in the bilayer, was further confirmed by solid-state NMR on liposomes containing $^{13}$C-labeled REM-CA peptides on three residues: L180 and G188 in R1 and I194 in R2 (underlined in *Figure 2A*). Solid-state NMR spectra confirmed that R1 adopted a single non-helical conformation in PC, while partially folding into an alpha helix in the presence of PIPs and sitosterol (*Figure 2G*, *Figure 2—figure supplement 3C*).

## Molecular dynamics simulations reveal interactions between REM-CA residues and lipids in the ternary lipid mixture

Molecular Dynamics (MD) simulation was performed with REM-CA and a bilayer composed of PC, sitosterol and PI4P, see *Video 1*. MD confirmed that REM-CA inserted itself in the lipid bilayer and presented two distinct regions (*Figure 2—figure supplement 4A*) in good agreement with the in silico analyses (*Figure 2—figure supplement 2*). MD proposed that albeit facing the inside of the bilayer (*Figure 2—figure supplement 4B*), the lateral ring of tyrosine Y184 (tyrosine being a residue often observed in interaction with sterols [*Nasir and Besson, 2011*]) was unlikely to interact with sterols with a distance between Y184 and sterol superior to 1 μm (*Figure 2H*). MD also modeled that lysines and arginine present in REM-CA, namely K192 and K193, and to a lesser extent K183 and R185, can form salt-bridges with the phosphate groups of PI4P (*Figure 2H*, *Figure 2—figure supplement 4B*). The lysine K176 was not in interaction with PI4P.

Altogether, we propose a model for the structure of REM-CA inserted in the PM inner-leaflet, composed of two domains: a PI4P-mediated alpha-helical folding conformation for R1 arranged on the PM surface interacting with the lipid polar heads through lysines and arginine and a hydrophobic conformation for R2 embedded inside the lipid phase (*Figure 2I*).

Classically, protein interactions with lipids occur through TM segments (*Lin and London, 2013*) or for monotopic proteins (to which REMs belong) through GPI anchoring, amphipathic helixes or ionic interactions (*Vinothkumar and Henderson, 2010*; *Hedger et al., 2015*). Moreover, specific interactions with PIPs usually occur through well-described motifs such as PH, C2 or PDZ domains (*Di Paolo and De Camilli, 2006*). The membrane-anchoring properties of REM-CA that we reveal here are therefore unconventional: these properties do not fit into any of the aforementioned lipid-interacting patterns and to the best of our knowledge such a membrane-anchoring conformation is not described in structural databases at present.

## Positively charged residues of REMORIN C-terminal Anchor are essential for PM-targeting

To further test the role of REM-CA residues found by MD in putative interactions with lipids (*Figure 2H*), we followed a near-iterative approach by observing the in vivo localization of YFP-*St*REM1.3 REM-CA mutants (*Figure 3A,B*). First, Y184 was mutated to a phenylalanine. Consistent with a lack of interaction with sterols (*Figure 2H*), the YFP-*St*REM1.3 $^{Y184F}$ mutant was still organized in PM nanodomains (*Figure 4A*). Second, we observed the subcellular localization of 19 YFP-*St*REM1.3 single to sextuple substitution mutants of the four lysines and the arginine present in REM-CA. Confocal microscopy images presented in *Figure 3C* show that single and double mutants still localized to the PM. In contrast, a strong impairment in PM-targeting with full or partial localization in the cytosol was observed for all triple to sextuple mutants. These results confirm the involvement of electrostatic interactions between REM-CA and negatively charged lipids with regard to PM targeting. Interestingly, in contrast to membrane surface-charge targeted proteins which generally possess a net charge of up to +8 (*Simon et al., 2016*; *Heo et al., 2006*; *Yeung et al., 2008*; *Barbosa et al., 2014*), the net electrostatic charge of REM-CA mutants is negative (*Figure 3B*). This suggests that the REM-CA/PM coupling is controlled by a specific lipid-peptide interaction, primarily governed by the intrinsic structural properties of the REM-CA moiety, and not by its net charge.

## Positively charged residues of REMORIN C-terminal anchor are essential for PM-nanodomain identity and function

We next focused on the mutants that still were targeted to the PM to test whether the mutations had an effect on their localizations in nanodomains. Consistently with MD calculations of the distance with the PI4P polar-head (*Figure 2H*), mutating K192, K193, and K183 revealed the requirement of

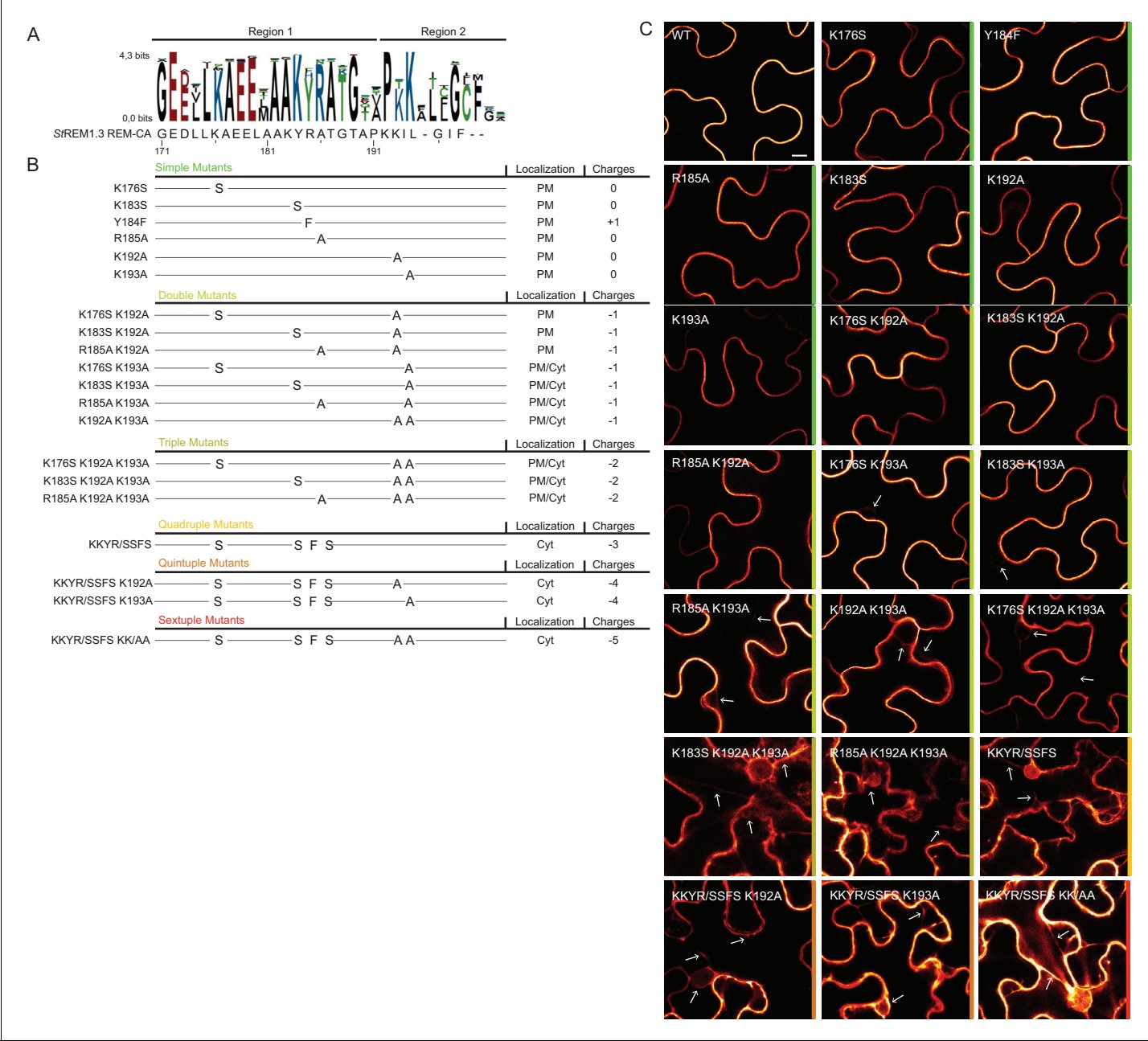

**Figure 3.** Positively charged residues of REMORIN C-terminal anchor are essential for PM targeting. (**A**) Sequence Logo obtained from 51 Group 1 REM-CA sequences presented in *Figure 1—figure supplement 1*, and *St*REM1.3 REM-CA sequence. (**B**) Summary of the 20 REM-CA mutants of *St*REM1.3 generated in this study and their corresponding subcellular localizations. PM, Plasma Membrane; Cyt, Cytosol. The total electrostatic charge of each mutated REM-CA is indicated. (**C**) Confocal images presenting secant views of *N. benthamiana* epidermal cells expressing 20 YFP-*St*REM1.3 REM-CA mutants (single to sextuple mutations), 48 hr after agroinfiltration. Scale bar of 10μm applies to all images.

these residues for a correct nanodomain organization whereas K176 and R185 taken alone are dispensable. Moreover, the coupling of K183, K192 and K193 mutations with other mutations on charged residues increased the alteration of *St*REM1.3 PM-nanodomain organization as assessed by the SCI (*Figure 4A,B*).

To address the functional relevance of REM nanodomain-organization, we exploited the previously reported role of *St* REM1.3 in restricting cell-to-cell propagation of PVX by decreasing plasmodesmata size-exclusion limit ( *Raffaele et al., 2009* ; *Perraki et al., 2014* , *Perraki et al., 2012* ).

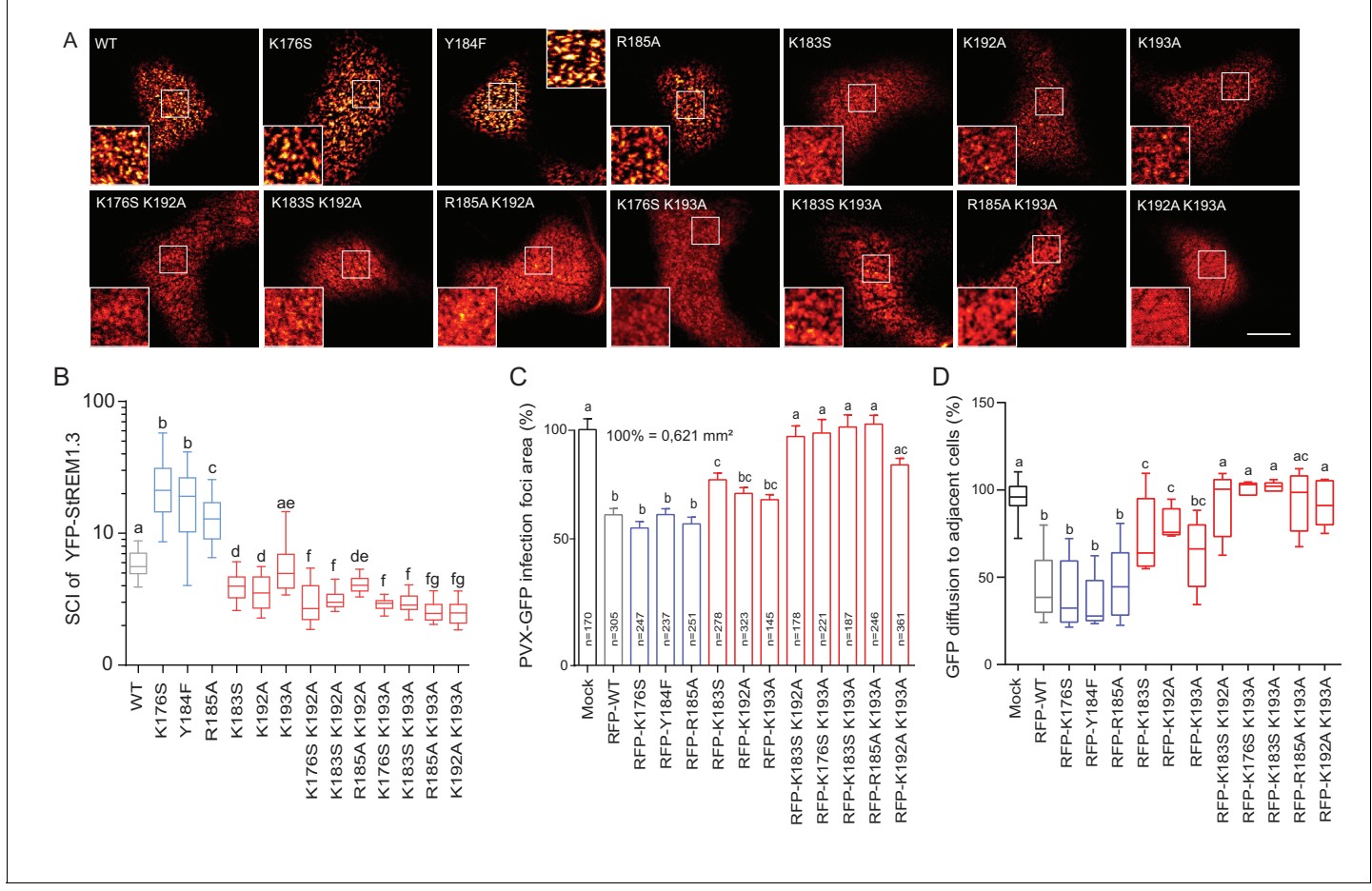

**Figure 4.** Positively charged residues of REMORIN C-terminal Anchor are essential for PM nanodomain localization and REMORIN function in cell-to-cell permeability. ( **A** ) Surface view confocal images of the localization of REM-CA single and double mutants. Scale bar, 10 μm . ( **B** ) Tukey boxplot showing the Spatial Clustering Index of the REM-CA single and double mutants. Letters indicate significant differences revealed by Dunn's multiple comparisons test p < 0.05 ( n = 3 ). ( **C** ) Quantification of *Potato Virus X* fused to GFP (PVX: GFP) cell-to-cell movement alone (Mock) or co-expressed with *St* REM1.3 WT or *St*REM1.3 REM-CA single and double mutants. Tukey boxplots represent the PVX: GFP infection foci area normalized to the mock condition. Letters indicate significant differences revealed by Dunn's multiple comparisons test p < 0.05 ( n = 3 ). ( **D** ) Plasmodesmal permeability assessed in the presence of WT, single or double mutants of REM-CA, according to ( *Perraki et al., 2012* ). Tukey boxplots represent the percentage of cells presenting a free diffusion of the GFP ( n = 3 ), letter indicate significant differences revealed by Dunn's multiple comparisons test p < 0.05 (Statistical analysis in *Figure 4—figure supplement 1* ).

The online version of this article includes the following figure supplement(s) for figure 4:

**Figure supplement 1.** Effect of *St* REM1.3 REM-CA mutant over-expression on plasmodesmata permeability and PVX cell-to-cell movement.

Single mutants K176S, Y184F and R185A behaved like *St* REM1.3 [WT] whereas K183S and K192A and K193A partially lost their ability to reduce viral intercellular movement and PD permeability ( *Figure 4C, D* and *Figure 4—figure supplement 1*). A close to complete loss of activity was observed with REM-CA double mutants, unequivocally linking the protein lateral segregation with its function to regulate cell-to-cell connectivity.

## Single-particle tracking localization microscopy reveals that REMORIN C-terminal anchor mutants display a lower diffusion coefficient mobility

To better characterize the PM-localization of REM-CA mutants, we used single-particle tracking photoactivated localization microscopy in variable angle epifluorescence microscopy mode (spt-PALM VAEM [ *Manley et al., 2008* ]), *Video 2* . This super-resolution microscopy technique allows the reconstruction of high-density super-resolved nanoscale maps of individual protein localization and trajectories in the PM ( *Hosy et al., 2015* ). Different kinetic and organizational parameters, such as

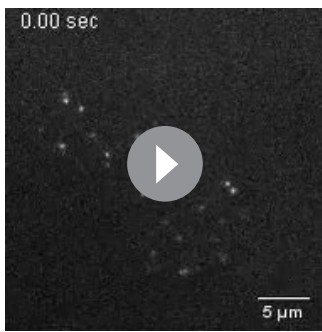

**Video 2.** Live-cell single-particle tracking-photoactivable localization microscopy in variable angle epifluorescence microscopy mode. RAW DATA spt-PALM VAEM was performed on *N. benthamiana* leaf epidermal cells expressing EOS-*St* REM1.3. https://elifesciences.org/articles/26404#video2

individual diffusion coefficient (D), mean square displacement ( MSD), nanodomain diameter and protein density can be calculated. We selected four REM-CA mutants that located at the PM but showed impairment in both nanodomain clustering and biological functions, namely StREM1.3 $^{K183S}$, StREM1.3 $^{K192A}$, StREM1.3 $^{K183S / K192A}$ and StREM1.3 $^{K192A / K193A}$ ( *Figure 5A –C* ). All EOS- *St* REM1.3 fusions exhibited a typically highly confined diffusion mode, but the four EOS-StREM1.3 mutants show a lower mobility than the EOS-StREM1.3 $^{WT}$ ( *Figure 5B, C* ). Nevertheless, EOS- *St* REM1.3 $^{K183S}$ displayed a similar MSD than EOS- *St* REM1.3 $^{WT}$ whereas EOS-*St* REM1.3 $^{K192A}$, EOS- *St* REM1.3 $^{K183S / K192A}$ and EOS- *St* REM1.3 $^{K192A / K193A}$ displayed a lower MSD ( *Figure 5D* ). *Figure 5E* depicts

**Figure 5.** REMORIN C-terminal anchor defines protein mobility in the PM. ( **A** ) Super-resolved trajectories ( trajectories > 20 points ;) of EOS- *St* REM1.3 WT and REM-CA mutants (K183S, K192A, K183S / K192A and K192A / K193A) visualized by high-resolution microscopy spt-PALM VAEM . 3. ( **B, C** ) Distribution of diffusion coefficients ( D ) represented as log (D) of the different fusion proteins and distributions of the peak D values of individual cells obtained by normal fits and were plotted as log (D), bar indicates mean ± sem ( **D** ) Mean Square Displacement ( MSD ) over time for the global trajectories > 15 pointsof each EOS- *St* REM1.3 construct ( n = 13 to 51 cells over three independent experiments). ( **E** ) Representative trajectories of *St* REM1.3 WT and REM-CA mutants. Significant differences revealed by Dunn's multiple comparisons.

representative trajectories of EOS-StREM1.3 [WT] and REM-CA-mutants.

## Live PALM data reveals that REMORIN C-terminal anchor defines protein segregation

To describe the supra-molecular organization of the proteins at PM level, we next analyzed live PALM data using Voronoï tessellation (*Levet et al., 2015*). This method subdivides a super-resolution image into polygons based on molecules local densities (*Figure 6A*, see online methods). For all fusion proteins, we identified clusters and precisely computed their dimensions (*Figure 6A,B*). EOS-*St*REM1.3 [WT] clustered in nanodomains with a mean diameter of *ca.* 80 nm, a result in good agreement with previous studies using different methods of imagery (*Raffaele et al., 2009*; *Demir et al., 2013*). For the WT protein fusion, nanodomains represented *ca.* 7% of the total PM surface (*Figure 6C*) with *ca.* 37% of molecules in nanodomains (*Figure 6D*) and a density of *ca.* two nanodomains per μm (*Raffaele et al., 2009*) (*Figure 6E*). Interestingly, the four REM-CA mutants showed a decrease of the total surface occupied by nanodomains in the total PM surface (*Figure 6C*). EOS-*St*REM1.3[K183S] and StREM1.3[K183S/K192A] display smaller nanodomains with a lower number of molecules per cluster, whereas EOS-*St*REM1.3[K192A], and EOS-*St*REM1.3[K192A/K193A] displayed larger nanodomains with a decrease of overall nanodomain density in the PM (*Figure 6D,E*). The study of REM-CA mutants revealed that single protein mobility behavior and protein supramolecular organization are uncoupled, for example EOS-*St*REM1.3[K192A] proteins displaying the lower MSD but forms larger clusters.

Altogether, spt-PALM and live PALM data analyzes showed that mutations in REM-CA affect the mobility and the organization of the protein by altering the partition of *St* REM1.3 molecules into nanodomains ( *Figures 3–6* ), likely causing the functional impairments observed. These results can be discussed in view of in silico spatial simulations of signaling events suggesting that proper partition of proteins optimizes signaling at PM ( *Mugler et al., 2013* ). In the case of *St* REM1.3, an altered partition in nanodomain is sufficient to inhibit the signaling events involving *St* REM1.3 in the

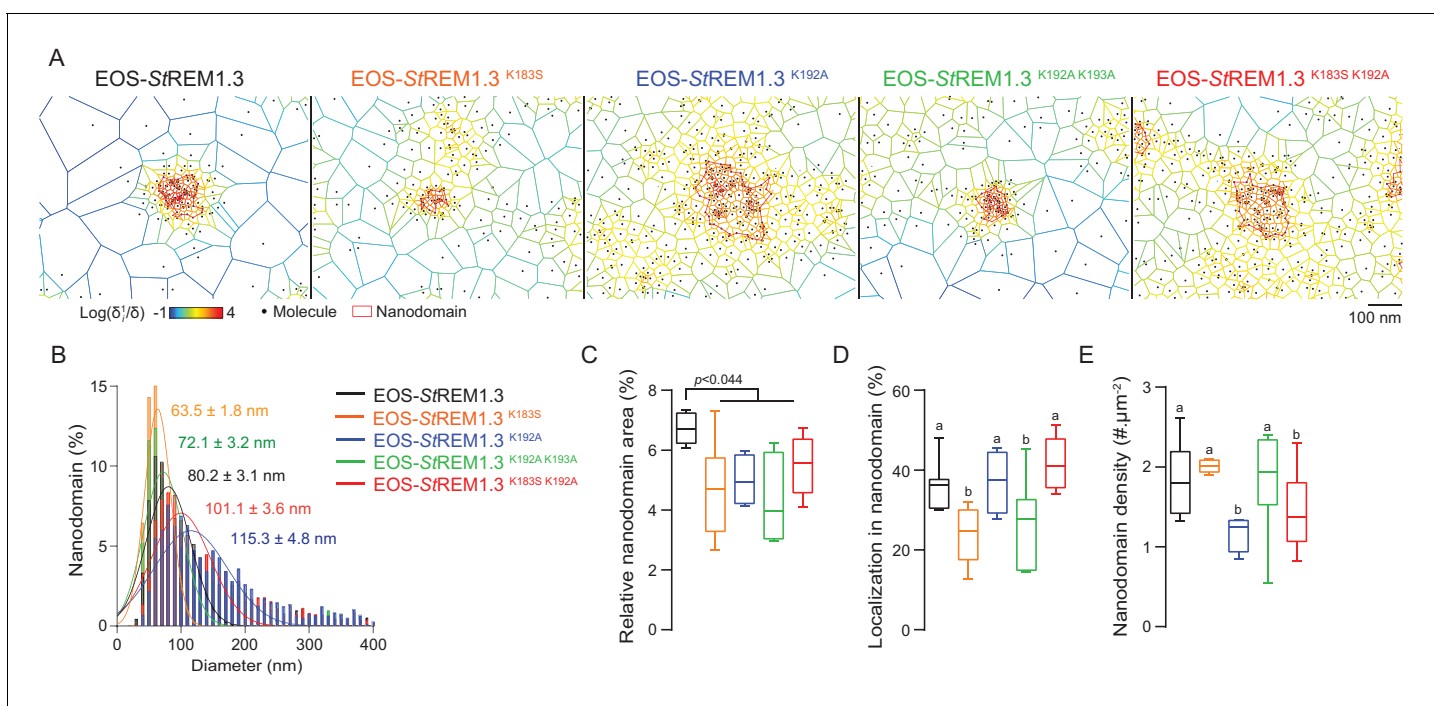

**Figure 6.** REMORIN C-terminal anchor defines protein segregation in nanodomains. ( **A** ) Live PALM analysis of molecules localization by tessellation-based automatic segmentation of super-resolution images. ( **B** ) Diameter distributions of the cluster of EOS fusion proteins (line shows the Gaussian fit). ( **C** ) Nanodomains surface expressed in percentage of the total PM surface. ( **D** ) Percentage of molecule localizing in nanodomain. ( **E** ) Nanodomain density (number of nanodomain.μm$^{-2}$) for the indicated proteins. Statistics were performed on at least six data sets per construction, see online methods for details. Letter indicate significant differences revealed by Dunn's multiple comparisons test p <0.05 .

PM. *St*REM1.3 being a phosphorylated protein, one can hypothesize that the REM-CA mutations alter the partition with its unknown cognate kinase (s) and / or interacting partners. Moreover, *St* REM1.3 locates to both the PM and in plasmodesmata , REM-CA mutations may also alter the partition between these two PM sub-compartments. These hypotheses are currently investigated in our laboratories.

## Conclusions

### Interaction between REM-CA and lipids mediates plant PM-nanodomain organization

Plant and animal plasma membranes are dynamically sub-compartmentalized into membrane domains ( *Lingwood and Simons, 2010* ). In plants, the REMORIN protein family represents the best-studied PM-domain-associated proteins ( *Jarsch and Ott, 2011* ). Genetic, live cell imaging and biochemical studies suggest that REM domains segregate into molecular platforms involved in hormone signaling and plant-microbe interactions ( *Jarsch and Ott, 2011* ; *Gui et al., 2016* ), but the functional relevance of REM PM-nanodomain organization and the molecular mechanisms underlying PM-nanodomain organization of REM are still unknown.

Here, we provide an unprecedented resolution of the molecular mechanisms that control protein spatio-temporal segregation into membrane nanodomains. Our work reveals that the group 1 REM *St* REM1.3 is targeted to inner-leaflet PM nanodomains from the cytosol by REM-CA, an unconventional C-terminal structural lipid-binding motif that undergoes a conformational change in presence of PI4P and sterols ( *Figures 1* and *2* ). These lipids seem to form the core components modulating REM nanodomain organization. The precise cooperativity between each lipid in the inner-leaflet remains to be determined ( *Vonkova et al., 2015* ), in particular the role of sterols, which do not seem to be in direct interaction with REM-CA ( *Figures 2* and *3*). It is well established that sterols interact preferentially with saturated-fatty acid containing lipids to form highly ordered lipid phases ( *Lorent and Levental, 2015* ). Interestingly, plant PM PI4P contain up to 30 - 60 % of saturated acyl chains ( *Furt et al., 2010* ; *König et al., 2007* ), we may assume that this preferential interaction is one of the driving force that allow the clustering REM -enriched domains ( *Figure 1* ). Following our model, a dynamic famille-à-trois between REM-CA, PI4P and sterols inside the inner-leaflet phospholipid bilayer PM inner-leaflet could lead to a definition of what drives PM-nanodomain formation ( *Figure 2* ).

### REM-CA diversity and REM-CA-like moieties beyond the plant kingdom

Mutations of *St* REM1.3 REM-CA residues involved in the interaction with the polar heads of phosphoinositides modify PM lateral segregation and dynamics and consequently the protein's functionality ( *Figures 4*, *5* and *6* ). This reflects the importance of REM-CA in the definition of *St* REM1.3's lateral segregation and it shows that lateral segregation is determined by the primary sequence of REM-CA. The six different phylogenetic groups of REMs label spatially distinct PM-domains ( *Jarsch et al., 2014* ; *Raffaele et al., 2007* ). It follows that the evolution of the REM-CA sequence could be involved in the diversification of the different PM-domains marked by REMs (*Raffaele et al., 2007* ). A more in-depth analysis of REM-CAs from other groups and the involvement of lipids in domain localization of REMs will provide crucial information about the determinants of PM lateral organization of the REM protein family that could *in fine* allow the deciphering of critical PM- associated signaling events in plants. In addition, it is reasonable to think that the S-acylation of REMs on their REM-CA moieties could modify the protein-lipid interaction that modulate their dynamics ( *Hemsley et al., 2013* ) ( *Figure 1—figure supplement 1* ). Study of a prenylated K-Ras protein in mammalian cells showed a complex structural cross-talk between the primary sequence of the protein and its prenyl moeity ( *Zhou et al., 2017*). Therefore, the study of acylated REM-CAs could provide another level of complexity in the establishment of REM-associated PM-nanodomains.

The structural conformation of REM-CA is original and does not fit with other membrane-anchoring conformation already described in databases. The search for structural analogues of REM-CA in publicly available structure databases identified analogous domains in bacteria, viruses and animals ( *Raffaele et al., 2013* ). Thus, the understanding of the structural basis for REM-CA PM-binding and lateral segregation may bring about knowledge of crucial importance, spanning beyond the plant kingdom.

## PM nanodomains represent a functional unit for plant cell signaling

In plants, most membrane proteins are relatively immobile forming static membrane domains ( *Jarsch et al., 2014* ; *Hosy et al., 2015* ; *Martinière et al., 2012* ). It was therefore postulated that membrane domain formation and functionality are based on protein immobility. Unexpectedly, we show in the present work that non-functional REM-CA mutants showing an altered PM-nanodomain localization harbor an even lower diffusion coefficient than the WT ( *Figure 5* ). This observation reveals that PM-protein function does not rely solely on their immobility but rather on their ability to organize into supramolecular domains. In good agreement, REM-CA mutants show an altered ability to partition into nanodomains ( *Figure 6*). It thus appears that REM-CA-defined PM-nanodomains may represent a functional unit for membrane-bound cell signaling in plants. The study of REM-CA mutants also reveals that the mobility behavior of single molecules is not directly linked to the partitioning of the resulting population. For example EOS- *St* REM1.3 $^{K192A}$ proteins displaying a lower MSD compared to the WT but form larger clusters. Similarly, a higher MSD is not necessarily coupled to a higher diffusion coefficient as observed for EOS- *St* REM1.3 $^{K183S}$ ( *Figure 5* ).

Altogether, our data reveal an unsuspected complexity in the definition of molecule organization and dynamics in the PM that we hope will pave the way for an exhaustive comprehension of the mechanisms regulating membrane-bound protein organization and function.

## Materials and methods

### Online methods

### Plant material, culture and transformation

*Nicotiana benthamiana* plants were cultivated in controlled conditions ( 16 hr photoperiod, 25 ˚ C ). Proteins were transiently expressed *via Agrobacterium tumefaciens* as previously described in ( *Perraki et al., 2012* ). For subcellular localization studies and biochemical purification, plants were analyzed 2 days after infiltration using 0.2 OD agrobacterium suspension. For PVX: GFP spreading assays and gating experiments, plants were observed 5 days after infiltration. The *A. tumefaciens* GV3101 strain was cultured at 30 ˚ Con appropriate selective medium depending on constructs carried. For phosphoinositide homeostasis modulation, effects of phosphoinositide phosphatase expression were observed 20 - 24 hr post-infiltration ( *Simon et al., 2016* ).

### Treatment with Brefeldin A

Leaves transiently expressing each construct 48 hr post-agroinfiltration were infiltrated with a dH $_2$ O solution of Brefeldin A at a concentration of 50 µg / mL (B7651 SIGMA); from a DMSO stock solution; for 3 hr before observation. Mock conditions contain the same volume of DMSO alone. Leaves were then observed with a Zeiss LSM 880 confocal fluorescence microscope with an oil-immersion 63x lense using the appropriate excitation wavelengths for each fluorescent fusion proteins.

### Cloning, molecular constructs and peptides

All constructs were generated using either classical or 3-in-1 Gateway cloning strategies (www.life-technologies.com) with pDONR P4-P1R, pDONRP2R-P3, pDONR211 and pDONR207 as entry vectors, and pK7WGY2 (*Karimi et al., 2002*), pUBN-Dest::EosFP (*Grefen et al., 2010*) and pB7m34GW (*Karimi et al., 2007*) as destination vectors. *St*REM1.3 mutants were generated by site-directed mutagenesis as previously described in (*Taton et al., 2000*) with minor modifications. All constructs were propagated using the NEB10 *E.coli* strain (New England Biolabs). Ultrapure

REM-CA peptides were obtained by de novo peptide synthesis, Purity >98% with acetylation at the N-terminal (GenScript HK Limited).

### Viral spreading and gating assays

Viral spreading of PVX:GFP in *N. benthamiana* leaves was assessed as described in (*Perraki et al., 2012*) with some modifications: spreading of PVX:GFP was visualized by epifluorescence microscopy (using GFP long pass filter on a Nikon Eclipse E800 with x4 objective coupled to a Coolsnap HQ2 camera) at 5 days post-infection and the areas of at least 30 of PVX:GFP infection foci per condition and per experiment were measured using a custom made macro on ImageJ.

## Epidermal cells live imaging and quantification

Live imaging was performed using a Leica SP5 confocal laser scanning microscopy system (Leica, Wetzlar, Germany) equipped with Argon, DPSS and He-Ne lasers and hybrid detectors. *N. benthamiana* leaf samples were gently transferred between a glass slide and a cover slip in a drop of water. YFP and mCitrine (cYFP) fluorescence were observed with similar settings (i.e. excitation wavelengths of 488 nm and emission wavelengths of 490 to 550 nm). In order to obtain quantitative data, experiments were performed using strictly identical confocal acquisition parameters (e.g. laser power, gain, zoom factor, resolution, and emission wavelengths reception), with detector settings optimized for low background and no pixel saturation. Pseudo-colored images were obtained using the 'Red hot' look-up-table (LUT) of Fiji software (http://www.fiji.sc/). All quantifications were performed on raw images for at least min of 10 cells, at least two plants by condition with at least three independent replicates.

For quantification of the PM Spatial Clustering Index (SCI), which reveals the degree of segregation of fluorescence signal on the surface plane of the PM (*Figure 1—figure supplement 4*), fluorescence intensity was plotted with a 10 µm line on raw images of cells PM surface view, three line plots were randomly recorded per cell and at least 15 cells per experiments were analyzed. For each plot, the Spatial Clustering Index was calculated by dividing the mean of the 5% highest values by the mean of 5% lowest values. For fluorescence intensities quantification, the mean grey value was recorded using a region of interest (ROI) of 5 µm x 5 µm on PM surface view raw images.

## Confocal multispectral microscopy

di-4-ANEPPDHQ-labelled leaves were observed as described in (*Gerbeau-Pissot et al., 2014*) with a Leica TCS SP2-AOBS laser scanning microscope (Leica Microsystems, Germany) and a HCPL Apochromat CS 63x (N.A. 1.40) oil immersion objective. Images were excited with the 458 nm line and the 488 nm line of an argon laser for CFP and di-4-ANEPPDHQ respectively as described in (*Gerbeau-Pissot et al., 2014*). Fluorescence emissions were filtered between 465 and 500 nm for CFP. For di-4-ANEPPDHQ, to obtain ratiometric images, we recorded green and red fluorescence between 540 to 560 nm and 650 to 670 nm, respectively. The mean red/green ratio of pixels (RGM) corresponding to either the global membrane, the 10%, the 5% or the 2% of the most intense CFP pixels were compared on each image.

## spt-PALM VAEM, single molecule localization and tracking

N. *N. benthamiana* epidermal cells were imaged at room temperature. Samples of leaves of 2 week-old plants transiently expressing EOS-tagged constructs were mounted between a glass slide and a cover slip in a drop of water to avoid dehydration. Acquisitions were done on an inverted motorized microscope Nikon Ti Eclipse (Nikon France S.A.S., Champigny-sur-Marne, France) equipped with a 100× oil-immersion PL-APO objective (NA = 1.49), a Total Internal Reflection Fluorescence Microscopy (TIRF) arm, a Perfect Focus System (PFS) allowing long acquisition in oblique illumination mode, and a sensitive Evolve EMCCD camera (Photometrics, Tucson, USA), see *Video 2*. Images acquisitions and processing were done as previously described by *Hosy et al., 2015*).

SR-Tesseler software was used to produce voronoï diagrams, and subsequently quantify molecule organization parameters as previously recommended (*Levet et al., 2015*). Taking in account fluorophore photophysical parameters, localization accuracy and the first rank of local density of fluorescent molecules, correction for multiple detections occurring in a vicinity of space ($\omega$) and blinking tolerance time interval ($\tau$) are identified as the same molecule, merged together and replaced by a new detection at a location corresponding to their barycentre. Because first rank of local density of fluorescent molecules was below 0.5 mol/$\mu m^2$ (c.a ranking from 0.1 to 0.3 mol/$\mu m^2$), we used a fixed search radius $\omega$ of 48 nm as recommended (*Levet et al., 2015*). To determine the correct time interval $\tau$, the photophysics of the fluorophore namely the off-time, number of blinks per molecule and on-time distributions are computed for each cell. For example, for a dataset composed of 618,502 localizations, the average number of blinks per molecule was 1.42, and the number of molecules after cleaning was 315,929. As a control, the number of emission bursts (439,331), counted with $\tau = 0$, divided by the average number of blinks per molecule (1.42) was only 2.15% different. After correction for artefacts due to multiple single-molecule localization, we computed potential cluster using a threshold $\delta_{1i} > 2\delta_N$, where $\delta_N$ is the average localization density at PM level and $\delta_{1i}$ is the

density in presumed protein-forming nanocluster, with a minimal area of 32 nm$^2$ and with at least five localization by cluster.

## Coarse-grained molecular dynamics

Coarse-grained simulations have been carried out by using Gromacs v4.5.4 (*Hess et al., 2008*) on a 6-processor core i7 cluster. Coarse-graining reduces the complexity of the molecular system and is widely used to study peptide- or protein-membrane interactions (*Lindahl and Sansom, 2008*). The initial structures of REM-CA peptides have been modeled as all-atom α-helices using standard backbone angles (φ = −90° and ψ = −45°) and side chain conformers with the ribosome v1.0 software (*Crowet et al., 2012*). These models were converted to a CG representation suitable for the MARTINI 2.1 forcefield (*Crowet et al., 2012*) with the Martinize script and the coarse-grained peptide was placed in a simulation box at least at 1 nm from a pre-equilibrated PLPC bilayer of 128 lipids or a PLPC:Cholesterol:PIP (98:19:2) bilayer of 124 lipids (*Marrink et al., 2008*; *López et al., 2009*, *López et al., 2013*). The N- and C-terminal ends of the peptide are charged and a helical secondary structure topology is maintained between residues 172–187. With Martini forcefield, secondary structures have to be restrained and a rational a priori on the structure has to be made. In our case, the peptide representation as two domains is based on in silico analyses and on the FTIR and NMR data (see *Figure 2—figure supplements 2*, *3*). Water particles were then added as well as ions to neutralize the system. A 5000-steps steepest-descent energy minimization was performed to remove any steric clashes. Five 2.5 μs simulations have been run for each peptide. Temperature and pressure were coupled at 300 K and one bars using the weak coupling Berendsen algorithm (*Crowet et al., 2012*) with τ T = 1 ps and τ p=0.5 ps. Pressure was coupled semi-isotropically in XY and Z. Non-bonded interactions were computed up to 1.2 nm with the shift method. Electrostatics were treated with ε = 15. The compressibility was 105 (1/bars).

## In silico analysis of REM-CA from *St*REM1.3

Sequence and predicted structure of the StREM1.3 REM-CA peptide predicted by different methods (NPSA (https://npsa-prabi.ibcp.fr/cgi-bin/npsa_automat.pl?page=/NPSA/npsa_seccons.html), hydrophobic cluster analysis (HCA) 15, the consensus secondary structure prediction is indicated: alpha helix (h), random coil (c), beta sheet (e). In HCA plot of the StREM1.3 REM-CA sequence, V, F, W, Y, M, L and I are hydrophobic residues. These amino acids are circled and hatched to form hydrophobic clusters.

## Atomistic molecular dynamics

From the coarse-grained simulations, ending frames from one of the replicates have been taken to carry atomistic simulations. The conversion has been carried out as described in *Wassenaar et al., 2013*). Briefly, atomistic lipid fragments and amino acid are backmapped and the system is relaxed through several energy minimizations and molecular dynamic simulations with position restraints. Cholesterol is reversed to sitosterol (*Poger and Mark, 2013*) and PIP to PI4P (*Holdbrook et al., 2010*). Simulations have been performed with the GROMOS96 54a7 force field (*Schmid et al., 2011*) with the Berger topology for PLPC (*Poger et al., 2010*; *Poger and Mark, 2010*). All the systems studied were first minimized by steepest descent for 5000 steps. Then NVT and NPT equilibrations were carried on for 0.1 and 1 ns. The peptide was under position restraints and periodic boundary conditions (PBC) were used with a two fs time step. Production runs were performed for 50 ns. All the systems were solvated with SPC water (*Berendsen et al., 1981*), and the dynamics were carried out in the NPT conditions (300 K and 1 bar). Temperature was maintained by using the v-rescale method (*Bussi et al., 2007*) with τT = 2.0 ps and a semiisotropic pressure was maintained by using the Berendsen barostat (*Berendsen et al., 1984*) with a compressibility of 4.5 × 105 (1/bar) and τp=1 ps. Electrostatic interactions were treated by using the particle mesh Ewald (PME) method (*Darden et al., 1993*). Van der Waals and electrostatics were treated with a 1.0 nm cut-off. Bond lengths were maintained with the LINCS algorithm (*Hess et al., 1997*). The trajectories were performed and analysed with the GROMACS 4.5.4 tools as well as with homemade scripts and softwares, and 3D structures were analyzed with both PYMOL (DeLano Scientific, http://www.PyMOL.org) and VMD softwares (*Humphrey et al., 1996*).

## TLC analysis of phosphoinositides mix (PIPs)

Phosphoinositides mix (PIPs), #P6023 SIGMA, was separated by HP-TLC plate along with authentic standards: Phosphatidylserine (PS) Phosphatidylinositol (PI), Phosphatidylinositol-4-phosphate (PI4P) and Phosphatidylinositol-4,5-bisphosphate PI(4,5)P2) (*Furt et al., 2010*). HP-TLC were stained with Primulin and relative amounts of each lipid species present in the PIPs mix quantified by densitometry scanning (*Macala et al., 1983*)

## Fourier transformed-infrared (FTIR) spectroscopy

MLV were prepared by rehydrating the resulting films with $D_2O$ or Tris-HCl buffer (10 mM pH 7.0) for FT-IR experiments as described previously (*Nasir et al., 2016*). Lipids were co-dissolved in chloroform/methanol (2:1, v/v) without or with peptides at a 10-to-1, lipid-to-peptide molar ratio. FTIR spectra of lipid-peptide MLV were recorded on Bruker Equinox 55 spectrometer (Karlsruhe, Germany) equipped with a liquid nitrogen-cooled Deuterated Triglycine Sulfate (DTGS) detector. The spectra were measured with a spectral resolution of 4 $cm^{-1}$ and are an average of 128 scans. All the experiments were performed with a demountable cell (Bruker) equipped with CaF2 windows (*Nasir et al., 2013*). During the experiments, the spectrophotometer was continuously purged with filtered dry air. MLV solution containing or not peptides was deposited into the $CaF_2$ window-equipped cell. All FTIR spectra were representative of at least two independent measurements. The attribution of different peaks was carried out according to the literature (*Arrondo and Goñi, 1998*; *Kong and Yu, 2007*).

## Adsorption experiments at constant surface area

Peptide adsorption into lipid monolayer was recorded on an automated Langmuir film system (KSV Minitrough 7.5 × 20 cm, Biolin Scientific, Stockholm, Sweden). The lipid monolayers (1-Palmitoyl-2-linoleyl-*sn*-glycerol-3-phosphocholine (PLPC) alone, PLPC-Sitosterol (80–20 molar ratio) or PLPC-sitosterol-phosphoinositides sodium salt from bovine brain (P6023 SIGMA, see *Figure 2—figure supplement 1B*), 70-20-10 molar ratio, were formed by spreading a precise volume of lipid solutions prepared in chloroform/methanol (2:1 v/v). After stabilization of the lipid monolayer at a defined initial surface pressure, peptides (solubilized Tris-HCl, 10 mM pH7.0 22 ± 1°C) were injected in the sub-phase (Tris-HCl, 10mM pH7.0, 22 ± 1°C) to a final concentration of 0.16 μM. The surface pressure variation is recorded over time. Experiments at different initial surface pressures were performed in order to plot the maximal surface pressure increase ($\pi_{max}$) as a function of the initial surface pressure ($\pi_i$) and to determine the maximal insertion pressure (MIP) as previously described (*Eeman et al., 2009*; *Deleu et al., 2014*; *Nasir and Besson, 2011*).

## Sample preparation for NMR

To prepare multilamellar vesicles (MLV), REM-CA peptides were solubilised in chloroform/methanol (2:1, v:v) and mixed with the appropriate amount of lipid powder (DMPC, PIPs and sitosterol) adjusting the REM-CA-to-lipid ratio (1:25). Solvent was evaporated under $N_2$ airflow to obtain a thin lipid film. Lipids were rehydrated with ultrapure water before lyophilisation. The lyophilized powder was then hydrated with appropriate amount of deuterium depleted water and homogenized by three cycles of shaking in a vortex mixer, freezing (liquid nitrogen, −196°C, 1 min) and thawing (40°C in a water bath, 10 min). This protocol leads to a milky suspension of micrometer-sized MLVs.

## Solid-state NMR spectroscopy

$^2$H NMR experiments were carried out on Bruker Avance III 400 MHz (9.4 T) and Bruker Avance III 500 MHz (11.75 T) spectrometer. Samples were equilibrated 30 min at a given temperature before data acquisition. $^2$H NMR experiments on $^2$H-labeled DMPC were performed at 76 MHz with a phase-cycled quadrupolar echo pulse sequence (90°x-τ−90°y-τ-acq). $^{31}$P NMR spectra were acquired at 202 MHz using a phase-cycled Hahn-echo pulse sequence (90°x-τ−180°x/y-τ-acq). Acquisition parameters were set as follows: spectral window of 50 kHz for $^{31}$P NMR, 250 kHz for $^2$H NMR, π/2 pulse widths of 15 μs for $^{31}$P and 2.62 μs for $^2$H, interpulse delays τ were of 40 μs, recycle delays ranged from 1.1 to 5 s. 2 k to 4 k scans were used for $^2$H and $^{31}$P NMR experiments, depending on the sample. The spectra were processed using Lorentzian line-broadening of 100 to 200 Hz for $^2$H spectra.

$^{13}$C experiments were recorded on a Bruker Avance III 800 MHz (18.8 T) at 11 kHz magic angle spinning (MAS) frequency. Sample temperature was held constant at −12°C with the internal reference DSS (*Böckmann et al., 2009*). The two dimensional proton-driven spin diffusion $^{13}$C-$^{13}$C (PDSD) spectra were recorded with an initial $^{1}$H-$^{13}$C cross-polarization and a mixing time of 50 ms. Acquisition times were set to 8 ms and 20 ms in the indirect and direct dimension, respectively, and the interscan delay was chosen to 2 s. Proton decoupling during acquisition with a frequency of 90 kHz was applied, using the SPINAL-64 decoupling sequence (*Fung et al., 2000*). All the spectra were processed and analyzed using Bruker Topspin 3.2 software and the Ccpnmr Analysis software (*Vranken et al., 2005*).

## Statistics
For all statistical analyses, ANOVA and Tukey's honestly significant difference tests were performed with Graphpad Prism, in order to find means that are significantly different from each other. Box-plots were drawn using Graphpad Prism (horizontal bars in the boxes represent the median, boxes the interquartile range, whiskers extend out 1.5 times the interquartile range, and individual points are outliers), and other graphs were drawn using excel software (Microsoft, https://products.office.com/).

## Acknowledgements

Imaging was performed at the Bordeaux Imaging Center, member of the national infrastructure France BioImaging. JG and PG are supported by the Ministère de l'Enseignement Supérieur et de la Recherche, France (MERS, doctoral grants). This work was supported by the Bordeaux Metabolome Facility-MetaboHUB (grant no. ANR–11–INBS–0010). We acknowledge the platform Métabolome-Fluxome-Lipidome of Bordeaux (http://www.biomemb.cnrs.fr/INDEX.html) for contribution to lipid analysis. BH and SM thank the Appel conjoint IdEx Bordeaux – CNRS PEPS 'REM solid' for funding. LL and MD thank the FNRS (PDR grant T.1003.14), the Belgian Program on Interuniversity Attraction Poles initiated by the Federal Office for Scientific, Technical and Cultural Affairs (IAP P7/44 iPros), the University of Liège (Fonds Spéciaux de la Recherche, Action de Recherche Concertée-Project FIELD) for financial support. MD and LL are Senior Research Associates for the Fonds National de la Recherche Scientifique (FRS-FNRS), JMC is supported by the IAP iPros project and MNN by the ARC FIELD project. Partial computational resources for MD simulations are provided by the Belgian 'Consortium des Équipements de Calcul Intensif' (CECI), funded by the FRS-FNRS under Grant No. 2.5020.11. YJ is funded by ERC no. 3363360-APPL and Marie Curie Action, no. PCIG-GA-2011–303601 under FP/2007–2013. We acknowledge the support of ANR (13-PDOC-0017–01 to BH, ANR-14-CE09-0020-01 to AL and Project CONNECT ANR-14-CE19-0006-01 to EMB), CNRS and the University of Bordeaux (IdEx - Chaire d'Installation and PEPS) to BH, the FP7 program (FP7-PEOPLE-2013-CIG to AL) and the ERC (ERC Starting Grant to AL, agreement no. 105945). SR is supported by the European Research Council (ERC-StG-336808) and the French Laboratory of Excellence project 'TULIP' (ANR-10-LABX-41; ANR-11-IDEX-0002–02). We thank Yohann Boutté, Amélie Bernard and Patrick Moreau for the critical reading of the manuscript. We acknowledge Patrick Moreau for the gift of SAR1 mutant clone, Véronique Santoni for the gift of aquaporin AtPIP1.1 and Coralie Chesseron for greenhouse facilities. No conflict of interest declared.

## Additional information

### Funding

| Funder | Grant reference number | Author |
| --- | --- | --- |
| Labex | French Laboratory of Excellence TULIP ANR-10-LABX-41 | Jean-Marc Crowet |
| ARC FIELD project | | Jean-Marc Crowet |
| Fonds De La Recherche Scientifique - FNRS | 2.5020.11 | Jean-Marc Crowet |

| Labex | French Laboratory of Excellence TULIP ANR-11-IDEX-0002-02 | Jean-Marc Crowet |
|---|---|---|
| IdEx Chaire d'excellence | | Birgit Habenstein |
| Agence Nationale de la Recherche | 13-PDOC-0017-01 | Birgit Habenstein |
| Centre National de la Recherche Scientifique | PEPS project Idex Bordeaux - PEPS 2316 | Birgit Habenstein Sébastien Mongrand |
| European Research Council | ERC starting grant 336808 | Sylvain Raffaele |
| European Research Council | ERC starting grant ERC-StG-105945 | Antoine Loquet |
| Agence Nationale de la Recherche | ANR-14-CE09-0020-01 | Antoine Loquet |
| Agence Nationale de la Recherche | Project CONNECT, ANR-14-CE19-0006-01 | Emmanuelle M Bayer |
| European Research Council | ERC starting grant 3363360-APPL | Yvon Jaillais |
| Marie Curie Action | PCIG-GA-2011-303601 FP/2007-2013 | Yvon Jaillais |
| Agence Nationale de la Recherche | ANR-11-INBS-0010 | Véronique Germain Sébastien Mongrand |
| Fonds De La Recherche Scientifique - FNRS | T.1003.14 | Laurence Lins |
| Federaal Wetenschapsbeleid | Belgian Program of Interuniversity Attraction Poles IAP P7/44 iPros | Laurence Lins |

The funders had no role in study design, data collection and interpretation, or the decision to submit the work for publication.

## Author contributions

Julien Gronnier, Conceptualization, Resources, Data curation, Software, Formal analysis, Validation, Visualization, Methodology, Writing—original draft, Writing—review and editing; Jean-Marc Crowet, Conceptualization, Software, Formal analysis, Visualization, Methodology, Writing—original draft; Birgit Habenstein, Conceptualization, Formal analysis, Supervision, Funding acquisition, Validation, Investigation, Methodology, Writing—original draft; Mehmet Nail Nasir, Conceptualization, Formal analysis, Methodology, Writing—original draft; Vincent Bayle, Matthieu Pierre Platre, Resources, Data curation; Eric Hosy, Conceptualization, Software; Paul Gouguet, Data curation, Writing—original draft; Sylvain Raffaele, Resources, Validation; Denis Martinez, Data curation, Software, Formal analysis; Axelle Grelard, Software, Formal analysis; Antoine Loquet, Conceptualization, Resources, Validation, Writing—original draft; Françoise Simon-Plas, Emmanuelle M Bayer, Resources, Methodology; Patricia Gerbeau-Pissot, Resources, Data curation, Formal analysis, Methodology; Christophe Der, Data curation, Formal analysis; Yvon Jaillais, Conceptualization, Resources, Formal analysis, Writing—original draft; Magali Deleu, Conceptualization, Resources, Formal analysis, Methodology; Véronique Germain, Conceptualization, Resources; Laurence Lins, Conceptualization, Data curation, Software, Formal analysis, Supervision, Funding acquisition, Investigation, Visualization, Methodology, Writing—original draft; Sébastien Mongrand, Conceptualization, Resources, Data curation, Formal analysis, Supervision, Funding acquisition, Validation, Investigation, Methodology, Writing—original draft, Project administration, Writing—review and editing

## Author ORCIDs

Mehmet Nail Nasir ![ORCID] http://orcid.org/0000-0003-3429-9445
Sylvain Raffaele ![ORCID] http://orcid.org/0000-0002-2442-9632
Sébastien Mongrand ![ORCID] https://orcid.org/0000-0002-9198-015X

Decision letter and Author response
Decision letter https://doi.org/10.7554/eLife.26404.sa1
Author response https://doi.org/10.7554/eLife.26404.sa2

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
