## [Decision Letter]

Thank you for submitting your article "Structural Basis for Plant Plasma Membrane Protein Dynamics and Organization into Functional Nanodomains" for consideration by *eLife*. Your article has been reviewed by three peer reviewers, and the evaluation has been overseen by a Reviewing Editor and Christian Hardtke as the Senior Editor.

The reviewers have discussed the reviews with one another and the Reviewing Editor has drafted this decision to help you prepare a revised submission.

Plant plasma membranes are organized into membrane domains that are implicated in mediating specific functionalities. Gronnier et al. have employed a variety of methods to provide mechanism for microdomain targeting of the REMORIN proteins in living plant cells. Remorins are considered markers for plant plasma membrane "nanodomains". Overall, the results from this study represent a clear step forward in the understanding of plant plasma membrane sub-compartmentalization and will/should be of interest to a broad audience interested in molecular mechanisms by which various physiological processes are controlled at the plasma membrane. The authors use an impressive array of different techniques, including different methods for high-resolution imaging, biophysical assays on lipid-protein interactions of purified recombinant proteins, lipid analysis and molecular dynamics modeling. The broad range of complementary approaches represents a clear strength of the work. A key piece of information is the characterization of the c-terminal anchor of remorins, REM-CA, to propose an interesting new mechanism by which remorins might be not only peripherally attached to membranes, but are actually deeply embedding themselves into the membrane bilayer, following interaction with the headgroups of anionic membrane phospholipids.

Summary:

All reviewers were impressed by the application of a variety of complimentary methods to address the experimental problem. However, there was some feeling that the paper was a bit too biophysical and more could be done to relate the findings to the overall biology. There were also concerns about specific claims that appear to go beyond the data presented. There was also concerns noted with regard to providing the proper citations and the perhaps overuse of citations to your own past work. The more technical criticisms are noted below.

Essential revisions:

1) The authors make a point that PI4P is am major determinant of the behavior of the plasma membrane/cytosol interface (Simon et al., Nat Plants 2, 16089, 2016). They are using SAC1 to modulate the membrane levels of PI4P, which is demonstrated nicely using different fluorescent markers. The (quite convincing) data are somewhat in conflict with the previous report that SAC1 is a PI(3,5)P2 5-phosphatase (Zhong et al., Plant Cell 17, 1449-1466, 2005). To clear this important point, I suggest to further support the effect of SAC1 expression specifically on PI4P by quantitative lipid analysis.

2) To support the biophysical data, the molecular dynamics modeling is initiated to support the notion that REM-CA interacts with PI4P, resulting in the insertion of the protein into the lipid bilayer. However, in the simulation not PI4P is used but PI(4,5)P2, which might behave quite different from PI4P and has not been implicated in mediating PI4P-like large-scale effects on the inner leaflet of the plasma membrane (Simon et al., Nat Plants 2, 16089, 2016). The reason for using a PI(4,5)P2 model is weak ("… because PI4P was not available in lipid structural databases for MD…") and the molecular dynamics simulation should be repeated using the correct lipid PI4P. It would in fact be quite interesting to see whether or not the simulation with either lipid might give different results, adding another layer of understanding.

3) The molecular dynamics simulation as presented is based on coarse-grain modeling, reducing the complexity of the interacting partners to relatively few spheres of biophysical influence. The authors should provide the rationale for choosing the particular degree of simplification and better state the limitations arising from their coarse-grain approach.

4) The weak point of this manuscript is the lack of functional data in endogenous condition, instead all experiments in plant cells have been carried out by transiently overexpressing heterologous proteins (potato) (different constructs) in heterologous system, tobacco cells. Thus, the presented results will be compelling if proven that those proteins are functional in tobacco cells at the expression levels studied. In addition, the conclusion "PM nanodomains represent a functional unit for plant cell signaling" is not supported by data in this manuscript. Note that we do not necessarily expect you to address this point by additional experimentation but would like to see some caveats added to the paper pointing out these weaknesses and how that may or may not affect the conclusions reached.

---

## [Author Response]

Summary:All reviewers were impressed by the application of a variety of complimentary methods to address the experimental problem. However, there was some feeling that the paper was a bit too biophysical and more could be done to relate the findings to the overall biology. There were also concerns about specific claims that appear to go beyond the data presented. There was also concerns noted with regard to providing the proper citations and the perhaps overuse of citations to your own past work. The more technical criticisms are noted below.

We are also very pleased to see acknowledged the importance of our study for a broad audience. We also thank the reviewers for the positive and constructive comments and for acknowledging the amount of various methods we used to understand a molecular mechanism regulating protein sub-compartmentalization in the plant plasma membrane.

We have simplified parts of the text, notably the parts related to the biophysical assays in order to go straight to the results and we removed unnecessary details that are already present in the Materials and methods section. We carefully reviewed the citations to be sure that we acknowledged all the communities working in the different fields related to our work and we hope that these new citations will adequately respond to your concerns.

Essential revisions:1) The authors make a point that PI4P is am major determinant of the behavior of the plasma membrane/cytosol interface (Simon et al., Nat Plants 2, 16089, 2016). They are using SAC1 to modulate the membrane levels of PI4P, which is demonstrated nicely using different fluorescent markers. The (quite convincing) data are somewhat in conflict with the previous report that SAC1 is a PI(3,5)P2 5-phosphatase (Zhong et al., Plant Cell 17, 1449-1466, 2005). To clear this important point, I suggest to further support the effect of SAC1 expression specifically on PI4P by quantitative lipid analysis.

The SAC1 construct used in our study is derived from the yeast *Saccharomyces cerevisiae* (further called SAC1p described in (1), and not from *Arabidopsis thaliana*, further called AtSAC1. We wish to apologize for the imprecision.

Yeast SAC1p displays a specific PI4P phosphatase activity well documented in vitro (2 3 4 5), as well as in vivo (1) in yeast and mammalian cells. This specificity is in good agreement with our controls presented in Figure 1—figure supplement 6 showing the depletion of PI4P in the PM by the MAP-fused SAC1p, and with previous results presented in Simon et al., Nat Plants 2016.

2) To support the biophysical data, the molecular dynamics modeling is initiated to support the notion that REM-CA interacts with PI4P, resulting in the insertion of the protein into the lipid bilayer. However, in the simulation not PI4P is used but PI(4,5)P2, which might behave quite different from PI4P and has not been implicated in mediating PI4P-like large-scale effects on the inner leaflet of the plasma membrane (Simon et al., Nat Plants 2, 16089, 2016). The reason for using a PI(4,5)P2 model is weak ("… because PI4P was not available in lipid structural databases for MD…") and the molecular dynamics simulation should be repeated using the correct lipid PI4P. It would in fact be quite interesting to see whether or not the simulation with either lipid might give different results, adding another layer of understanding.

To take the reviewer’s comment into account, molecular dynamic simulations have been repeated with the PIP4 lipid. The general model topology of PIP proposed for Martini forcefield has been used for the coarse-grained simulations and the atomistic topology has been built by modifying the PI(4,5)P2 topology. From the simulations, the same interaction and insertion observations between REM-CA and the ternary membrane composition have been made, i.e. on one hand, K192 and K193 form salt-bridges with the phosphate group of PI4P, and on the other hand the two regions of REM-CA have a different behavior, with a primary and deeper insertion of R2. The Figure 2 and Figure 2—figure supplement 4 have been modified to reflect the calculation results found for the new PI4P model.

3) The molecular dynamics simulation as presented is based on coarse-grain modeling, reducing the complexity of the interacting partners to relatively few spheres of biophysical influence. The authors should provide the rationale for choosing the particular degree of simplification and better state the limitations arising from their coarse-grain approach.

Coarse-grained (and coarse-grained to atomistic transformations) are a general and widely used approach to study peptide- or protein-membrane interactions, notably lipid-specific interactions (1, 2). This approach is used to speed up simulations and observe phenomenons that could not be observed by atomistic simulations. In particular, coarse-grained simulations have been recognized as useful for peptide membrane-insertion analysis (3). Preliminary atomistic simulations have shown LYS/ARG – PIP phosphate interactions but to simulate the complete insertion of REM-CA, several µs at least would have been required. This argument further reinforces the choice for coarse-grained simulations. Nevertheless, as pointed out by the reviewer, the Martini forcefield comes with limitations, for example secondary structures have to be restrained. In our case, the peptide representation as two domains is not only based on in silico analysis predicting R1 as having amphipathic and helical properties and R2 as having unstructured and more hydrophobic properties, but also on the FTIR and NMR experiments. We assumed this structural basis as sufficient and we used it for the coarse-grained modeling. The text has been amended to better present these points.

1) Stansfeld, P.J., Hopkinson, R., Ashcroft, F.M. & Sansom, M.S. PIP(2)-binding site in Kir channels: definition by multiscale biomolecular simulations. Biochemistry 48, 10926-10933 (2009).

2) Arnarez, C., Mazat, J.P., Elezgaray, J., Marrink, S.J. & Periole, X. Evidence for cardiolipin binding sites on the membrane-exposed surface of the cytochrome bc1. J Am Chem Soc 135, 3112-3120 (2013).

3) Lindahl, E. & Sansom, M.S. Membrane proteins: molecular dynamics simulations. Curr Opin Struct Biol 18, 425-431 (2008).

4) The weak point of this manuscript is the lack of functional data in endogenous condition, instead all experiments in plant cells have been carried out by transiently overexpressing heterologous proteins (potato) (different constructs) in heterologous system, tobacco cells. Thus, the presented results will be compelling if proven that those proteins are functional in tobacco cells at the expression levels studied.

We agree that working with stable lines expressing endogenous group 1 REMs under the control of their endogenous promoter is the most desirable premise to carry out this type of research. However, the generation of 29 stable lines for the REM-CA mutants that would be required for the localization experiments would have been very tedious and, as explained further, not necessarily significantly more informative. Group 1 REM are highly expressed genes and amongst the 10% most expressed in plants (6). This justifies the use of a strong promoter such as p35S for confocal and functional experiments, and pUB10 for spt-PALM. Moreover, Jarsch et al., 2014 (7) showed that REMORINs labelled the same microdomains when expressed either under their native or under the 35S promoter. In addition, we showed by spt-PALM that StREM1.3 labels nanodomains of 80 nm (Figure 6), a result in total agreement with Raffaele et al. 2009 (8) (nanodomain size: 70 nm measured by immunogold on highly purified PM vesicles) and Demir et al. 2013 (9) (nanodomain size: 90 nm measured by STED microscopy). In addition, all live-cell imaging of REM-CA mutants has been performed 2 days after agroinfiltration, a time frame for which expression levels are 40 to 60% weaker than in the commonly used time frame for agrobacterium-mediated transient-expression (As observed in Figure 2B of (10)). Finally, several recent excellent publications in the plant field have used heterologous transient-expression of tagged plant PM proteins to study their dynamic localization pattern using microscopy (e.g. 11, 12, 7, 13, 14) and most, if not all, similar studies in animals (all of them being published in high-ranking journals, e.g. Zhou et al. Cell 2017 (15) are also performed upon transient heterologous expression under the control of very strong synthetic promoter.

Nevertheless, to strengthen the used of our biological system, we performed additional experiments that are included in the revised version of the manuscript (Figure 1—figure supplement 2). We showed that the closest orthologs of StREM1.3 in N. benthamiana described in (16) named NbREM1.2 and NbREM1.3 following the proposed nomenclature in (6), are expressed in N. benthamiana epidermis, encode for PM nanodomain-localized proteins; and that NbREM1.2, NbREM1.3 and, StREM1.3 expressed at similar expression levels display an identical ability to reduce PVX infection foci size (Figure 1—figure supplement 2).

Taking into account that the Potato Virus X propagates from plant-to-plant by mechanical dissemination via the epidermis (17), our newly added results support the physiological relevance of the overall presented work.

In addition, the conclusion "PM nanodomains represent a functional unit for plant cell signaling" is not supported by data in this manuscript. Note that we do not necessarily expect you to address this point by additional experimentation but would like to see some caveats added to the paper pointing out these weaknesses and how that may or may not affect the conclusions reached.

We absolutely agree that we did not directly address the question of the functionality of plant PM-nanodomains in terms of signaling. Nevertheless, we provide, for the first time in leaves, direct evidence that protein nanodomain-organization and dynamics at the PM are critical for protein function. Thus, the mechanisms described in the manuscript seem crucial to the organization of the StREM1.3 protein into domains that are functional for our physiological outputs. In agreement with the reviewers, we modified the Discussion section in regard to this pertinent remark.

References:

1) Hammond, G.R. et al. PI4P and PI(4,5)P2 are essential but independent lipid determinants of membrane identity. Science 337, 727-730 (2012).

2) Stefan, C.J. et al. Osh proteins regulate phosphoinositide metabolism at ER-plasma membrane contact sites. Cell 144, 389-401 (2011).

3) Mesmin, B. et al. A four-step cycle driven by PI(4)P hydrolysis directs sterol/PI(4)P exchange by the ER-Golgi tether OSBP. Cell 155, 830-843 (2013).

4) Moser von Filseck, J., Vanni, S., Mesmin, B., Antonny, B. & Drin, G. A phosphatidylinositol-4-phosphate powered exchange mechanism to create a lipid gradient between membranes. Nat Commun 6, 6671 (2015).

5) Moser von Filseck, J. et al. INTRACELLULAR TRANSPORT. Phosphatidylserine transport by ORP/Osh proteins is driven by phosphatidylinositol 4-phosphate. Science 349, 432-436 (2015).

6) Raffaele, S., Mongrand, S., Gamas, P., Niebel, A. & Ott, T. Genome-wide annotation of remorins, a plant-specific protein family: evolutionary and functional perspectives. Plant physiology 145, 593-600 (2007).

7) Jarsch, I.K. et al. Plasma Membranes Are Subcompartmentalized into a Plethora of Coexisting and Diverse Microdomains in Arabidopsis and Nicotiana benthamiana. The Plant cell 26, 1698-1711 (2014).

8) Raffaele, S. et al. Remorin, a solanaceae protein resident in membrane rafts and plasmodesmata, impairs potato virus X movement. The Plant cell 21, 1541-1555 (2009).

9) Demir, F. et al. Arabidopsis nanodomain-delimited ABA signaling pathway regulates the anion channel SLAH3. Proceedings of the National Academy of Sciences of the United States of America 110, 8296-8301 (2013).

10) Grefen, C. et al. A ubiquitin-10 promoter-based vector set for fluorescent protein tagging facilitates temporal stability and native protein distribution in transient and stable expression studies. The Plant journal: for cell and molecular biology 64, 355-365 (2010).

11) Mbengue, M. et al. Clathrin-dependent endocytosis is required for immunity mediated by pattern recognition receptor kinases. Proceedings of the National Academy of Sciences of the United States of America 113, 11034-11039 (2016).

12) Martiniere, A. et al. Cell wall constrains lateral diffusion of plant plasma-membrane proteins. Proceedings of the National Academy of Sciences of the United States of America 109, 12805-12810 (2012).

13) Bozkurt, T.O. et al. Rerouting of plant late endocytic trafficking toward a pathogen interface. Traffic 16, 204-226 (2015).

14) Somssich, M. et al. Real-time dynamics of peptide ligand-dependent receptor complex formation in planta. Sci Signal 8, ra76 (2015).

15) Zhou, Y. et al. Lipid-Sorting Specificity Encoded in K-Ras Membrane Anchor Regulates Signal Output. Cell 168, 239-251 e216 (2017).

16) Bozkurt, TO et al. The Plant Membrane-Associated REMORIN1.3 Accumulates in Discrete Perihaustorial Domains and Enhances Susceptibility to Phytophthora infestans. Plant physiology 165, 1005-1018 (2014).

17) Cruz, SS, Roberts, AG, Prior, DA, Chapman, S. & Oparka, KJ Cell-to-cell and phloem-mediated transport of potato virus X. The role of virions. The Plant cell 10, 495-510 (1998).